# Overcoming Challenges in Pediatric Formulation with a Patient-Centric Design Approach: A Proof-of-Concept Study on the Design of an Oral Solution of a Bitter Drug

**DOI:** 10.3390/ph15111331

**Published:** 2022-10-27

**Authors:** John Dike N. Ogbonna, Edite Cunha, Anthony A. Attama, Kenneth C. Ofokansi, Helena Ferreira, Susana Pinto, Joana Gomes, Ítala M. G. Marx, António M. Peres, José Manuel Sousa Lobo, Isabel F. Almeida

**Affiliations:** 1Drug Delivery and Nanomedicines Research Group, Department of Pharmaceutics, University of Nigeria, Nsukka 410001, Nigeria; 2Laboratory of Pharmaceutical Technology, Department of Drug Sciences, Faculty of Pharmacy, University of Porto, 4050-313 Porto, Portugal; 3Department of Chemical Sciences, Faculty of Pharmacy, University of Porto, 4050-313 Porto, Portugal; 4LAQV-REQUIMTE, Faculty of Pharmacy, University of Porto, 4050-313 Porto, Portugal; 5UCIBIO, REQUIMTE, Laboratory of Microbiology, Department of Biological Sciences, Faculty of Pharmacy, University of Porto, 4050-313 Porto, Portugal; 6Pharmacy Department, Portuguese Oncology Institute of Porto—IPO Porto, 4200-072 Porto, Portugal; 7Centro de Investigação de Montanha (CIMO), Instituto Politécnico de Bragança, 5300-253 Bragança, Portugal; 8Laboratório Associado para a Sustentabilidade e Tecnologia em Regiões de Montanha (SusTEC), Instituto Politécnico de Bragança, Campus de Santa Apolónia, 5300-253 Bragança, Portugal; 9UCIBIO, REQUIMTE, Med Tech, Laboratory of Pharmaceutical Technology, Department of Drug Sciences, Faculty of Pharmacy, University of Porto, 4050-313 Porto, Portugal; 10Associate Laboratory i4HB—Institute for Health and Bioeconomy, Faculty of Pharmacy, University of Porto, 4050-313 Porto, Portugal

**Keywords:** pediatric medicines, oral formulations, bitterness masking, patient centric design, in use stability

## Abstract

Designing oral formulations for children is very challenging, especially considering their peculiarities and preferences. The choice of excipients, dosing volume and palatability are key issues of pediatric oral liquid medicines. The purpose of the present study is to develop an oral pediatric solution of a model bitter drug (ranitidine) following a patient centric design process which includes the definition of a target product profile (TPP). To conclude on the matching of the developed solution to TPP, its chemical and microbiological stability was analyzed over 30 days (stored at 4 °C and room temperature). Simulation of use was accomplished by removing a sample with a syringe every day. Taste masking was assessed by an electronic tongue. The developed formulation relied on a simple taste masking strategy consisting in a mixture of sweeteners (sodium saccharine and aspartame) and 0.1% sodium chloride, which allowed a higher bitterness masking effectiveness in comparison with simple syrup. The ranitidine solution was stable for 30 days stored at 4 °C. However, differences were noted between the stability protocols (unopened recipient and in-use stability) showing the contribution of the simulation of use to the formation of degradation products. Stock solution was subjected to acid and alkali hydrolysis, chemical oxidation, heat degradation and a photo degradation stability assessment. The developed pediatric solution matched the TPP in all dimensions, namely composition suitable for children, preparation and handling adapted to hospital pharmaceutical compounding and adequate stability and quality. According to the results, in-use stability protocols should be preferred in the stability evaluation of pediatric formulations.

## 1. Introduction

The most preferred route for the administration of medicines in children is by far the oral one. Liquid dosage forms are particularly useful to children <5 years of age owing to the ease in swallowing and dose adjustment [1,2]. Oral liquid formulations provide maximal dosing flexibility, making it possible to use a single formulation over a wide age range (including neonates). Additionally, the dose of the drug can be easily and conveniently adjusted by measuring a different volume.

Designing a pediatric oral formulation is challenging with respect to several issues such as the choice of excipients, dosing volume and palatability [3,4]. Oral pharmaceuticals are frequently unpalatable and regarded as an unpleasant experience to most of the population, especially children. Palatability, a property resulting from the ensemble of several organoleptic properties such as smell, taste, aftertaste and texture (i.e., mouthfeel), is determined to ensure acceptance by the patient and adequate medication adherence. The taste masking in pediatric medicines for oral use is often crucial to improve their palatability. To this end, several taste masking strategies can be used, and preferences of children regarding oral liquid formulations need to be considered [5,6,7].

With respect to the selection of excipients and vehicle, several considerations can be made. Excipients may lead to adverse reactions in children, especially when used to treat infants and neonates. Methylparaben is associated with allergic reactions, and propyl paraben beyond allergic reactions may interfere with the physiological development of reproductive organs in animal models [8]. The use of alcohol and sucrose as excipients in pediatric formulations is also not recommended [9]. Syrups are frequently used as vehicles for oral compounding formulations. However, due to their high content of sucrose, they should be avoided for pediatric patients suffering from diabetes and hereditary fructose intolerance. Similarly, in long-term treatments, sugar-free formulations should be preferred to preserve dental health once sucrose changes dental plaque pH, dissolving tooth enamel and contributing to dental caries. Vehicles that contain a high concentration of sucrose are also prone to the occurrence of crystallization.

With regard to dosing volume, large volume doses may be problematic for both children and caregivers. The optimal dose volume has been defined as ≤5 mL for children under 5 years old and ≤10 mL for children. Consequently, a higher dosage is useful to ensure a low volume of administration, although the bitter taste of the liquid can be enhanced, thus requiring an effective taste masking strategy.

To successfully design an oral medicine for pediatric use, a systematic approach like a patient-centric design process can be followed. The patient-centric drug product design process can be defined as the “process of identifying the comprehensive needs of individuals or the target patient population and utilizing the identified needs to design pharmaceutical drug products that provide the best overall benefit to risk profile for that target patient population over the intended duration of treatment” [10].

Regulatory agencies have addressed the relevancy of placing the patient in the center of pharmaceutical development. The EMA has issued guideline/reflection papers for pediatric and older populations, [11,12,13] while the FDA [14] has developed a series of guidance documents on patient-focused drug development, with the primary goal to better incorporate the patient’s voice in drug development and evaluation.

Three major factors are analyzed in PCDPD, namely patient, drug and drug product characteristics. This systematic approach integrates this insight, which is translated to a Target Product Profile (TPP) to drive the pharmaceutical product design process.

Little information is available regarding the application of PCDPD to the design of oral pediatric formulations.

The European Medicines Agency has issued a specific guideline related to the pharmaceutical development of medicines for pediatric use in 2013. This key reference [13] describes the regulatory expectations for a pediatric medicinal product design including the end-user acceptability to optimize therapeutic outcomes. Considering pediatric medicines development, it is a challenge to find one formulation appropriate for all age groups and thus the age-related physiological and behavioural growth and their influence on the pharmacokinetics/pharmacodynamics and medicine use should be taken into account. The aim should be to safely cover a wide age range with a single formulation.

To apply PCDPD to the design of an oral pediatric formulation for a bitter drug, several factors should be taken into account, related to the patient (swallowing difficulty, flavour preferences, choice of excipients and dosing volume) and drug product (coverage for different age groups, palatability and stability). Ranitidine was used in this work as a model bitter drug. Ranitidine HCl (Ra-HCl) is a histamine H_2_-receptor antagonist used to prevent and treat gastro-esophageal reflux, gastric or duodenal ulcers and erosive esophagitis due to side effects of drugs as corticosteroids both in adults and children [15]. Ra-HCl oral solutions are available as syrups or low dosage (up to 15 mg/mL) sugar-free solutions [16,17]. The main objective of the study is to design an oral pediatric solution of a model bitter drug (Ra-HCl) following a patient-centric design process addressing key challenges in the formulation of oral pediatric medicines, and confirming if the resulting solution matched the defined TPP.

## 2. Results

### 2.1. Taste Masking Effectiveness vs. Standard Vehicle

The potential taste masking effect of the developed oral formulation regarding the bitterness of the Ra-HCl was qualitatively evaluated using the lab-made potentiometric E-tongue. The device, coupled with chemometric tools, allowed for comparing the taste profile, namely the bitter and/or sweet sensations, of the formulation (with or without Ra-HCl), of the standard vehicle (syrup with or without Ra-HCl), and of an aqueous solution containing the same amount of Ra-HCl (to which correspond the high bitter intensity). Multivariate E-tongue classification models (LDA-SA-E-tongue models) were established enabling the full discrimination (100% of correct classifications for training and LOO-CV procedures) of:i.syrup, with and without Ra-HCl, and water with Ra-HCl (Figure 1A), using a classification discriminant model with two DFs, explaining 100% of the data variability, based on the potential signals (in Volts) of two sensors: DF1 = −1066 × S2:11 + 963 × S2:12; DF2 = −56 × S2:11 − 68 × S2:12;ii.oral formulation, with or without Ra-HCl, and water with Ra-HCl (Figure 1B), using a classification discriminant model with two DFs, explaining 100% of the data variability, based on the potential signals (in Volts) of two sensors: DF1 = 3235 × S2:9 − 904 × S2:13; DF2 = −170 × S2:9 + 235 × S2:13; and,iii.the five oral solutions, using a classification discriminant model which two first DFs explained ~100% of the data variability, and were based on the potential signals (in Volts) of two sensors: DF1 = −1745 × S1:11 − 684 × S2:5 + 2511 × S2:11; DF2 = −292 × S1:11 + 226 × S2:5 + 333 × S2:11.

The satisfactory discrimination observed may be tentatively attributed to the known capability of this lab-made E-tongue to differentiate the basic taste sensations as well as to quantitatively respond to solutions with different bitter intensities. Indeed, the discriminant results and the position (more negative or more positive) of the studied samples according to the first DF (i.e., DF1) (Figure 1A,B) allow inferring that signals generated by sensors S2:9 and S2:11 are more related with the sweet sensation and the signals of sensors S2:12 and S2:13 by the bitter sensation. On the other hand, it should be noted that although sensors S1:11 and S2:11 have the same composition, from the results obtained for the five oral solutions, it is clear that they present an opposite behavior (based on the negative/positive sign of the respective coefficients). This can be tentatively attributed to the slight differences of physical properties (transparency and porosity) of the membranes due to the drop-by-drop technique used for building the device that may result in inhomogeneous membranes.

Furthermore, the individual 2D-plots confirmed that simple syrup is positioned much closer to the Ra-HCL solution than to the placebo, which can be interpreted as poor effectiveness in masking Ra bitterness (Figure 1A). On the other hand, the developed formulation is almost in an equidistant position from the placebo and the Ra-HCL solution (Figure 1B). When analyzing the combined results from the E-tongue analysis, it can be concluded that the oral solution developed in this work showed a better masking effectiveness than simple syrup, since it is more distanced from the aqueous solution of Ra-HCl and closer to placebo (Figure 1C). Globally, the E-tongue output supports that the developed formulation has better palatability than Ra-HCl solution prepared with simple syrup. However, it should be emphasized that the taste masking effect of the formulation developed should be interpreted with caution, since the results were not supported by a human panel analysis.

### 2.2. Chemical and Microbiological Stability

The chemical and microbiological stability of three batches of Ra-HCl formulation was studied over 30 days. Two storage conditions were used (4 °C and RT) and the recipient was opened once daily to remove 1 mL and thus simulate real use conditions. Unopened recipients stored at 4 °C and RT were included for comparison.

Regarding organoleptic features and pH (Table 1), the solution presented good results across storage conditions and over the entire analysis period. A slight yellow color was observed after 30 days at RT. Yellowing was already described for aqueous ranitidine solutions stored at 25 °C [18].

The content of Ra-HCl was higher than 98% for all test conditions (Table 2), which confirm the drug stability in the presence of the excipients used and the solution pH, and ensures a beyond use date of at least 30 days. However, when analyzing the chromatograms, new peaks appeared after 30 days of storage with once daily removal, in comparison with the unopened recipient at both storage temperatures (Figure 2), but most notably after storage at room temperature. This points to the formation of degradation products of the components of the solution.

The analysis of samples obtained from degradation studies was performed by comparing chromatograms generated from samples subjected to forced degradation studies to those generated following analysis of a freshly prepared solution of ranitidine. The results of the forced degradation study are given in Appendix A. The results suggested that Ra-HCl is unstable when exposed to base, light, and oxidative stressed conditions. When the chromatograms were examined, only one peak was observed in the degradation in the acidic hydrolysis, while two peaks appeared in the case of alkaline hydrolysis. When ranitidine solution was exposed to H_2_O_2_, there were several degradation peaks in addition to the ranitidine peak. The ranitidine was also susceptible to photolysis, with a decrease of 9.15% of the concentration when exposed to UVA light for 1 h. The ranitidine did not show a decrease in concentration during when analyzed under drastic thermal conditions, being considered practically stable to these conditions.

The results showed that the proposed method was able to separate Ra-HCl from the degradation products in all stress conditions.

Microbiological results after 30-day storage were in accordance with the requirements for oral dosage forms. Total aerobic microbial count at 35 °C, and total combined yeasts/mould count at 22 °C were inferior to the limit of 10^3^ CFU/mL and 10^2^ CFU/mL, respectively. The absence of *Escherichia coli* in 1 mL of solution was also confirmed. These results confirm the efficacy of the preservative system used, even when the recipient was open daily throughout the storage period.

## 3. Discussion

The patient-centric design process has been regarded as a successful approach to promote satisfaction with treatment and thus medication adherence [10]. Designing drug products for children should take into account the patient age and its developmental physiology, the disease to be treated, the route of administration, the dosing regimen, the maximum duration of the treatment, the age-associated activities of children (e.g., school, nursery), the setting where the medicine is likely to be used (e.g., hospital or community), and the caregivers’ characteristics. Oral liquid dosage forms with a low dosing volume and rapidly dissolving mini-tablets are preferable in infants [2]. Dosing measurement is a critical issue in pediatric care and the use of dosing devices is encouraged. The use of syringes is a useful approach for liquid formulations but requires a very fluid formulation for ease of handling. The formulation proposed in this study does not have thickening agents and thus is very fluid. Besides the dosage form and drug, excipients should also be carefully evaluated during pharmaceutical development. For example, the use of preservatives in pediatric drug products can be troublesome. The preservative system used in this work proved to be effective in the pH range defined as optimal to ensure Ra-HCl chemical stability and is regarded as safe for pediatric use [19]. Sorbic acid/potassium sorbate show a very good risk-benefit-relationship when compared with other preservatives. The effectiveness of these preservatives is limited to a pH range 3.5 to 5.5 which, in this formulation, is not a problem once as referred Ra-HCl may suffer hydrolysis in aqueous solution in pH outside the range 4.5–5.5. Additionally, potassium sorbate is very soluble in aqueous systems, does not require any co-solvents, and is the safest alternative for pediatric patients with respect to its efficacy for compounded oral preparations with a pH of 3.5 to 5.5 [1].

The dose volume is a major consideration for the acceptability of a liquid formulation. Typical target dose volumes for pediatric liquid formulations are <5 mL for children under 5 years and <10 mL for children of 5 years and older [20]. Increasing the drug loading favors low volume administration and also allows for more flexible dose adjustment in the course of therapeutic treatment. It is worthy of note that the Ra-HCl content of the developed solution is higher than the similar formulations (sugar-free) reported in formularies [15]. Increasing the drug content has, however, the disadvantage of increasing the bitter taste and can affect drug stability.

Active pharmaceutical ingredients, such as Ra-HCl, are often extremely bitter. Several taste masking techniques can be used in oral medicines such as complexation with ion-exchange resins, microencapsulation, prodrugs, inclusion complexation, granulation, and multiple emulsion or gel formation [6,7]. Some of them are impracticable in hospital and pharmacy settings. Hence, sweeteners and flavouring agents are usually added as masking compounds, although they are only partially effective. Sugar-free oral formulations are also preferred for children since sucrose promotes dental caries and should be avoided in pediatric patients with diabetes and hereditary fructose intolerance [21]. Blends of sweeteners (like sodium saccharin and aspartame) have shown a synergic effect and reduced aftertaste [17], and thus the strategy to use two sweeteners was adopted to improve the bitterness masking efficacy. Of note, aspartame may be harmful in patients with phenylketonuria and contra-indicated in homozygous autosomal recessive patients [21] The intensity of the sweetness and bitter taste masking capacity can be enhanced by the addition of a sodium salt [18]. Sodium chloride has been shown to be a potent inhibitor of some bitter compounds [19,22,23] Sodium chloride was herein used in a concentration similar to physiologic fluids and represents a cheap and simple strategy to mask bitter compounds. Flavor selection is also an important component in the development of oral pediatric products. Using flavors acceptable for children is an important requirement. Whilst bubble gum is a flavor that is usually appreciated by children, their preferences are known to be influenced by individual experiences and culture [24,25]. Giving children the opportunity to select the flavor prior the preparation of the oral formulations could be explored as a relevant contributor to satisfaction with treatment. Acceptability of the optimized formulation herein developed should be further evaluated in the pediatric population.

The study of the stability of compounded formulations is critical to determine the beyond use date. Changes in color, reduction in potency, and microbial growth are some of the stability problems encountered in liquid dosage forms. Thus, stability testing should address both chemical and microbiological parameters. According to a systematic review, the chemical and physical instabilities as well as microbial growth on pediatric oral extemporaneous formulations are very rare in published experimental studies [24]. However, stability protocols may not always take into account the usage conditions. Regarding chemical and microbiological stability, the Ra-HCl solution herein developed satisfied the requirements of European Pharmacopoeia after 30 days of storage, even considering the simulation of once daily administration. However, when analyzing the chromatograms related to in-use stability, other peaks appear, pointing to the chemical degradation of components of the solution. The simulation of use is an important parameter in the stability protocol once opening the bottle and removing a sample with a syringe can lead to microbiological contamination, chemical loss, and the formation of degradation products due to the exposure to atmospheric oxygen, environmental contaminants and microbial invasion. This could be especially critical to children since small concentration changes can lead to lack of effectiveness, and toxic effects can arise due to the appearance of degradation products. Thus, it is highly recommended that in-use stability protocols are used for pediatric formulations.

Taking the results into consideration, the characteristics of the developed Ra-HCl oral formulation matched the TPP previously defined, according to a PCDPD strategy, in every dimension.

### Limitations

In this study, taste masking effectiveness was assessed with an E-tongue and thus the results should be interpreted with caution, since they were not supported by a human panel analysis. The taste masking effectiveness was assessed after preparation of the oral solution and thus taste changes over time might occur.

## 4. Materials and Methods

### 4.1. Chemicals and Materials

Chemicals used in the chromatographic analysis were of analytical reagent grade and used without any further purification. Water from a Milli-Q plus with a specific conductance less than 0.1 µS cm^−1^ was used throughout. All materials used in the ranitidine solution were of pharmaceutical grade. Ra-HCl purity standard was acquired from Honeywell Fluka (Seelze, Germany), sucrose was purchased from Fagron Iberica (Barcelona, Spain), sodium saccharin, sodium sorbate, sorbitol, citric acid, sorbic acid, sodium chloride and aspartame were purchased from Acofarma, (Barcelona, Spain), bubble gum flavor was purchased from Ministério dos Remédios, (Porto, Portugal) and sodium citrate was obtained from Sigma Aldrich (Buchs, Switzerland).

HPLC grade acetonitrile was obtained from Merck (Darmstad, Germany), and monopotassium phosphate and sodium hydroxide was purchased from Fluka (Madrid, Spain).

Taking into account the comments mentioned in the introduction, a TPP for an oral pediatric solution of ranitidine is defined as described in Table 3.

### 4.2. Composition of the Oral Pediatric Solution

The composition of the Ra-HCl solution is described in Table 4. The solution contains a synergic mixture of sweeteners (sodium saccharin and aspartame) and sodium chloride to help mask Ra-HCl bitterness, besides preservatives (sorbic acid), humectant (sorbitol, also contributing as a sweetener) and pH buffer (citric acid and sodium citrate). The amount of preservatives, humectant and buffers used were defined according to the usual levels in pediatric oral formulations. In particular, the amounts of sweeteners and sodium chloride were defined following exploratory taste tests conducted by an experienced hospital pharmacist.

The formulation was prepared using magnetic stirring at 30 °C until complete dissolution (under 30 min). Bubble gum flavor was then added in drops (six drops) and the volume was increased to 50 mL in an amber-colored volumetric flask.

Simple syrup (66.7 wt. %) was used for comparison with the optimized formulation using an E-tongue. The syrup was prepared by mixing sucrose in water at 30 °C until full dissolution followed by filtration with a Chardin paper filter. Ra-HCl was added under heating (30 °C) and mixed until fully dissolved.

### 4.3. Evaluation of Taste Masking Effectiveness vs. Standard Vehicle (Simple Syrup) Using E-Tongue

#### 4.3.1. E-Tongue Apparatus and Potentiometric Analysis of Ra-HCl Solutions

An electronic tongue has been applied to pharmaceutical products for the assessment of bitter taste masking effectiveness [25]. Herein, a lab-made potentiometric E-tongue multisensor device (Figure 3), comprising two cylindrical arrays [26] and a Ag/AgCl reference electrode (Crison, model 5241), was used to establish the potentiometric fingerprints of the developed oral formulation and of the syrup (with or without ranitidine), as well as of aqueous solutions containing ranitidine. The E-tongue device was connected to an Agilent Data Acquisition unit (model 34970A), which was controlled by Agilent BenchLink Data Logger software, as shown in Figure 3. Each array comprised 20 lipid polymeric cross-sensitive sensor membranes (40 sensors in total), each corresponding to a different mixture of an additive compound and a plasticizer, with the compositions described in Appendix A.

The type of lipid polymeric sensors and the compositions were chosen based on previous research that demonstrated their ability to discriminate aqueous solutions that mimic the basic taste sensations (i.e., acid, bitter, salty, sweet and umami) as well as to quantitatively respond to solutions containing different concentrations of bitter substances (e.g., quinine monohydrochloride or caffeine) [26,27] or sugars (e.g., glucose, fructose and/or sucrose) [7]. Although the underlying mechanisms are not known, lipid membranes can interact with several bitter and sweet compounds through electrostatic/hydrophobic interactions, including the establishment of hydrogen bonds between, for example, hydroxyl groups of the target molecules and the hydroxyl, carboxyl, amine, and/or phosphate groups of the lipid membranes. Simultaneously, the possible adsorption of some target molecules into the membrane induces a potential signal change associated either with the rise of the membranes’ final depolarization degree or to variations in the membrane surfaces’ electric dipole orientation or density charge, enabling the detection. The sensitivity of each sensor depends on the concentration of the charged lipids inside the membrane. On the other hand, the selectivity of these sensors is dependent on the surface hydrophobicity, i.e., a high lipid concentration promotes the hydrophilicity and, oppositely, high plasticizer concentrations make the surface more hydrophobic [28]. It should be highlighted that the used lipid sensor membranes have global cross-sensitivity and low selectivity, being not specific to any particular substance, in a similar way to the biological receptors of the human gustatory system [29].

For the experimental potentiometric assays, six independent samples (100 mL each) of each formulation containing 2.5% (*w*/*v*) of ranitidine (developed solution and syrup, without flavour), plus the respective placebos, were analyzed, as well as an aqueous solution with the same amount of ranitidine. The placebos were prepared the same way as the drug products. The solutions were analyzed, at ambient temperature (~20 °C), with the E-tongue, for 5-min, which allowed reaching a pseudo-equilibrium between the E-tongue non-specific lipid polymeric membranes and the microorganism of each solution. After each assay the E-tongue was washed with deionized water and, after four to five assays, it was immersed in an HCl aqueous solution (0.01 mol/L), to evaluate the signal repeatability, as well as to promote a more efficient cleaning of the sensor arrays. At the end of each day, the E-tongue was stored at room temperature and immersed in an HCl cleaning solution (0.01 mol/L), ensuring the integrity and functionality of the lipid polymeric membranes during long time-periods.

#### 4.3.2. Statistical Analysis

The E-tongue performance, as a taste sensor device, regarding the differentiation of the studied oral formulations, syrup and water containing Ra-HCl or the respective placebos, based on their bitter intensities, was evaluated. For that, a linear discriminant analysis (LDA) coupled with the meta-heuristic simulated annealing (SA) variable selection algorithm [30,31,32,33] was implemented. The SA algorithm was used to identify the best sub-set of non-redundant E-tongue sensors [34] which could provide information about the capability of the selected oral formulation to mask the bitterness of the Ra-HCl and to compare it with that of the syrup. The LDA performance was checked for the original grouped data (training) as well as for the leave-one-out cross-validation (LOO-CV) procedure, graphically for the former, based on the 2D plots of the two most significant discriminant functions (DF) and based on the sensitivity values (i.e., the percentage of samples correctly classified into the pre-established groups) for the latter. The LOO-CV is a CV variant usually implemented for small datasets that do not allow the establishing of an external dataset for validation. At each step the approach keeps one observation from the original dataset as the validation data, and the remaining observations are used as the training data for developing the LDA model. This procedure is repeated until all observations are used once as the validation data. The predictive performance of the LDA model is then assessed based on the correct classification percentage of the observation kept as the validation data.

### 4.4. Study of the Stability of the Developed Oral Solution

#### 4.4.1. Sample Preparation

Three batches of the selected Ra-HCl oral solution at the concentration of 25 mg/mL were prepared according to the method described in 2.2. The solutions were poured into 100 mL amber glass bottles (commonly used in hospital pharmacies) and sealed with a screw-on cap. Samples were stored at room temperature or at 4 °C for 30 days. Daily, 1 mL of each solution was removed, simulating the conditions of use in a hospital setting (in-use stability). One sample for each test condition remained sealed.

#### 4.4.2. Microbiological Stability

Representative samples of the formulation were analyzed at 30 days storage according to the methods described in European Pharmacopoeia 9.0. Briefly, the total count of microorganisms by membrane filtration and incubation in Tryptic Soy agar at 35 °C and Sabouraud agar at 22 °C and detection of Escherichia coli by broth enrichment in Tryptic Soy Broth at 35 °C, subculture in Gram negative selective broth at 44 °C, followed by culture in Gram negative selective agar.

#### 4.4.3. Chemical Stability Evaluation

At 0, 7, 15 and 30 days, 1 mL samples were withdrawn and analyzed for pH, appearance, colour, and assay of the Ra-HCl.

##### pH Determination

The pH of the Ra-HCl solutions was determined in triplicate using a pH meter (Basic 2°, Crison Instruments, (Chicago, IL, USA).

##### Ranitidine Assay

The Ra-HCl content was analyzed with the following HPLC methodology.

##### Chromatographic Conditions

The chromatographic analysis was carried out using an HPLC, model LC-4000, integrated system (Jasco, Japan) equipped with an AS-4050 autosampler, a PU-4180 pump and a MD-4010 multiwavelength diode array detector (DAD) of the same brand. The following fused-core stationary-phase chemistries were used: a Thermo Scientific Hypersil Gold C18 (150 mm × 2,1 mm, 3μm) column, thermostated at 35 °C. Detection was performed in near UV region, from 190–400 nm, and absorbencies at the wavelength of 230 nm were used for Ra-HCl quantification. Sample injection volume was 10 μL. A gradient solvent system of 2% acetonitrile (A) and 28% acetonitrile (B) was used as follows: 0 min 100% A; 15 min 100% (B); 16 min 100% (A); at a constant flow 1.5 mL/min., the HPLC detector was set to 230 nm. The method was performed in compliance with European Pharmacopoeia [35].

To establish a stability-indicating assay, forced degradation studies of Ra-HCl solutions were carried out by means of hydrolytic (acid and basic), oxidative, thermal, and photolytic conditions. The acid hydrolysis occurred with the addition of hydrochloride acid to the Ra-HCl solution, with the final concentration of 0.1 M, with a duration of 8 h. Alkaline hydrolysis was performed by the addition of NaOH 0.1 M, with a duration of 8 h. After the exposure period, the samples submitted to hydrolytic stresses were neutralized. Oxidative stress was performed by exposure Ra-HCl solution to a 3% v/v H2O2 at room temperature for 24 h. Thermal degradation occurred through the submission of Ra-HCl solution at 75 °C in a forced circulation oven for 24 h. Photolytic degradation occurred by exposure of Ra-HCl solution to 254 nm at 200 Watts. h/m2 UV light in a UV cabinet for 1 h. In all stress situations the final working concentration of Ra-HCl solutions was 0.07 mg/mL. All samples were filtered and submitted to chromatographic analysis.

##### Method Validation

Method validation is an important part in method development that involves the process of defining analytical requirements and confirming that the method under consideration has performance capabilities consistent with those requirements. The following parameters were evaluated: linearity, precision, detection limit (LD) and quantification limit (LQ). Linear calibration curve for standard Ra-HCl was constructed over six calibration levels, 0.03 mg/mL, 0.05 mg/mL, 0.07 mg/mL, 0.09 mg/mL, 0.11 mg/mL, 0.14 mg/mL, each injected in triplicate. The precision was evaluated in terms of repeatability (within-day relative standard deviation, R.S.D.) and in terms of intermediate precision (between-day R.S.D.) at the same concentration level in three non-consecutive days by the analysis of a 0.04 mg/mL standard Ra-HCl solution (results displayed in Table 5).

## 5. Conclusions

Despite the development of new drugs and new dosage forms, the formulation of liquid dosage forms for oral administration to suit the special needs of pediatric patients remains an important task in hospital/community pharmacies. Pediatric oral formulation design poses different challenges that should be collectively addressed. The oral pediatric ranitidine solution herein developed corresponded to the TPP previously defined, and thus the PCDPD strategy was successfully implemented. Sodium chloride improved the masking of ranitidine bitterness and represents a simple and cheap taste masking approach that could be of use for other drugs with similar or lower bitterness values than Ra-HCl.

The developed formulations were even more effective than simple syrup (standard vehicle for oral solutions) in masking Ra-HCl bitterness. Based on our results, we also recommend that in-use stability should be evaluated for pediatric formulations. PCDPD offers a holistic approach to customized treatment, thus representing an invaluable option to improve patient satisfaction in the pediatric population.

## Figures and Tables

**Figure 1 pharmaceuticals-15-01331-f001:**
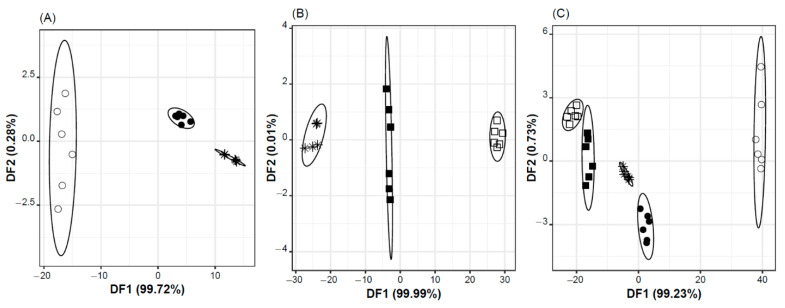
Supervised LDA discrimination of oral formulations, syrup and water with and without ranitidine (oral formulation without ranitidine (□); oral formulation with ranitidine (■); syrup without ranitidine (○); syrup with ranitidine (●); and/or, water with ranitidine (*)) based on non-redundant potentiometric signals of E-tongue sensors, selected by the SA algorithm. (**A**) LDA-SA-E-tongue model based on two E-tongue sensors (S2:11 and S2:12). (**B**) LDA-SA-E-tongue model based on two E-tongue sensors (S2:9 and S2:13). (**C**) LDA-SA-E-tongue model based on three E-tongue sensors (S1:11, S2:5 and S2:11).

**Figure 2 pharmaceuticals-15-01331-f002:**
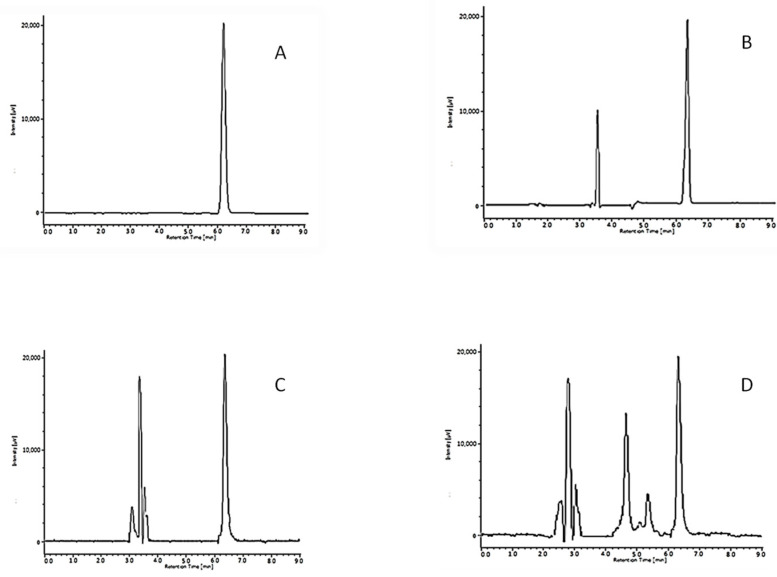
Chromatographic runs of Ra-HCl after 30 days preserved at 4 °C (**A**), after 30 days open daily preserved at 4 °C (**B**), after 30 days preserved at room temperature (**C**) and after 30 days open daily preserved at room temperature (**D**).

**Figure 3 pharmaceuticals-15-01331-f003:**
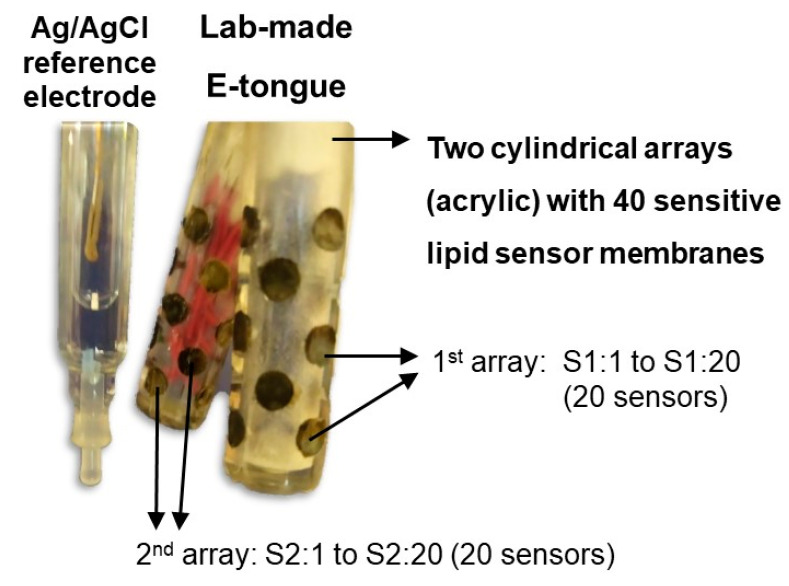
Lab-made E-tongue comprising two cylindrical sensor arrays, each with 20 lipid-polymeric sensor membranes (adapted with permission from Ghrissi et al. [26] ©MDPI, 2022).

**Table 1 pharmaceuticals-15-01331-t001:** Results of the stability evaluation of the oral pediatric ranitidine 25 mg/mL solution.

Time	Temperature	pH	Appearance	Color	Odor
0 days	4 °C	5.10 ± 0.03	clear	Colorless	Characteristic of bubble gum
	Room Temperature	5.12 ± 0.02	clear	Colorless	
7 days	4 °C	5.10 ± 0.02	clear	Colorless	
	Room Temperature	5.11 ± 0.02	clear	Colorless	
15 days	4 °C	5.09 ± 0.03	clear	Colorless	
	Room Temperature	5.15 ± 0.02	clear	Colorless	
30 days	4 °C	5.21 ± 0.01	clear	Colorless	
	Room Temperature	5.22 ± 0.01	clear	Slightly yellow	
30 days	Unopened 4 °C	5.20 ± 0.01	clear	Colorless	
30 days	Unopened Room Temperature	5.29 ± 0.01	clear	Slightly yellow	

**Table 2 pharmaceuticals-15-01331-t002:** Results of the chemical stability of the oral pediatric ranitidine 25 mg/mL solution. (Mean ± standard deviation, *n* = 3).

% of Initial Ranitidine Content			
Day	In-Use Stability (4 °C)	In-Use Stability (RT)	Unopened Recipient (4 °C)
0	100.0	100.0	100.0
7	99.8 ± 0.5	99.3 ± 0.4	
15	99.7 ± 0.2	99.0 ± 0.3	
30	99.5 ± 0.1	98.4 ± 0.7	99.7 ± 0.2

**Table 3 pharmaceuticals-15-01331-t003:** Target Product Profile (TPP) established for the oral pediatric solution of ranitidine.

Characteristic	Target	Comment
Concentration	25 mg/mL	Allows a low volume administration
pH	4.5–5.5	pH of maximum stability of Ra-HCl
Chemical and microbiological stability	At least 30 days	Comparison between solution in closed recipient and solution with once daily sample removal (in-use stability) [13]
Flavor	Bubble gum	One of the preferred flavors for children
Taste masking strategy	Sweeteners and sodium chloride	Synergistic effect of aspartame and sodium saccharine [17] and sorbitol Sodium chloride affords bitterness masking [18,19,20]
Viscosity	Very fluid	Appropriated for administration with a syringe
Excluded excipients	Sugar-free, alcohol-free and paraben free	Parabens and alcohol are not recommended for pediatric formulations [21]. Sugar-free formulation is non-cariogenic and suitable for children with diabetes and hereditary fructose intolerance [13]
Preparation	Simple, with low energy consumption	Feasible in a hospital pharmacy setting

**Table 4 pharmaceuticals-15-01331-t004:** Composition of the oral Ra-HCl 25 mg/mL solution (% *w*/*v*).

Sodium Saccharin	Sorbic Acid	Potassium Sorbate	Sorbitol 70%	Citric Acid	Aspartame	Sodium Chloride	Sodium Citrate
0.6	0.1	0.1	5	0.1	0.3	0.1	0.5

**Table 5 pharmaceuticals-15-01331-t005:** Analytical figures of merit of the method regarding calibration data, precision and sensitivity.

Calibration Equation	Determination Coefficient (R^2^)	LOD (mg/mL)	LOQ (mg/mL)	Repeatability (% RSD), *n* = 6	Intermediate Precision (% RSD), *n* = 6
y = 367.33x + 11.9	0.996	0.0075	0.0249	1.8	2.7

## Data Availability

Data are contained within the article and Appendix A.

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
