# Peer review of "Overcoming Challenges in Pediatric Formulation with a Patient-Centric Design Approach: A Proof-of-Concept Study on the Design of an Oral Solution of a Bitter Drug"

_pharmaceuticals, 2022, doi:10.3390/ph15111331_

Round 1

Reviewer 1 Report (Previous Reviewer 3)

I still disagree with the title that is totally misleading

It is at the most a proof-of-concept study with one ‘not so bitter’ API.

Also please check the references - -it is not uniform

-For some authors'  1st names are mentioned but not for others

- 17 is wrong

Allen L., P.-Jean E., Perello L, Hernandez C, et al, Stability of an oral ranitidine suspension (15 538

mg/ml)

it should be

Petit-Jean E, Perello L, Hernandez C, et alStability of an oral ranitidine suspension (15 mg/ml)European Journal of Hospital Pharmacy: Science and Practice 2013;20:46-49.

It makes me wonder about the others...

Author Response

I still disagree with the title that is totally misleading. It is at the most a proof-of-concept study with one ‘not so bitter’ API.

Our comment: We understand the point of view of the reviewer and change the title to: Overcoming challenges in pediatric formulation with a patient-centric design approach: a proof-of-concept study on the design of an oral solution of a bitter drug.

Regarding bitterness values, the bitterness of ranitidine is higher than paracetamol, urea and close to quinine, the reference compound for bitterness (Sarika Pradhan et al 2019 J. Phys.: Conf. Ser. 1380 012091.

 Also please check the references - -it is not uniform  -For some authors'  1st names are mentioned but not for others

- 17 is wrong Allen L., P.-Jean E., Perello L, Hernandez C, et al, Stability of an oral ranitidine suspension (15 mg/ml)  it should be

Petit-Jean E, Perello L, Hernandez C, et al. Stability of an oral ranitidine suspension (15 mg/ml)European Journal of Hospital Pharmacy: Science and Practice 2013;20:46-49.

Our comment: We thank the reviewer for the correction. The references were thoroughly revised.

Reviewer 2 Report (Previous Reviewer 1)

Authors have not addressed the following comments:

1. Taste masking, an important attribute of pediatric formulation, was not evaluated with stability samples. As some degradation of ranitidine was evident through chromatographic studies, taste may be compromised after one month of storage. Hence stability studies are required to be added.

2. Further study for identification of degradants seems essential to confirm the identity of degradants to assess safety of the developed formulation.

3. Limitations of the study have not been provided anywhere in the article.

Author Response

  1. Taste masking, an important attribute of pediatric formulation, was not evaluated with stability samples. As some degradation of ranitidine was evident through chromatographic studies, taste may be compromised after one month of storage. Hence stability studies are required to be added.

Our comment:  The reviewer raised a relevant issue. However, it is not feasible to repeat the whole study and for that reason information on this topic was added to the limitations section.

  1. Further study for identification of degradants seems essential to confirm the identity of degradants to assess safety of the developed formulation.

Our comment: Although we understand the point of view of the reviewer the analysis/discussion of the degradation products is outside the scope of the manuscript. This manuscript focused on strategies to overcome design challenges in pediatrics rather than a comprehensive characterization of the stability of ranitidine. Additionally, the main ranitidine degradation products are already described in pharmacopeia monographs.

  1. Limitations of the study have not been provided anywhere in the article.

Our comment: We agree with the reviewer. Limitations were now included in the manuscript.

In this study, taste masking effectiveness was assessed with an E-tongue and thus the results should be interpreted with caution, since they were not supported by a human panel analysis. Taste masking effectiveness was assessed after preparation of the oral solution and thus taste changes over time might occur.

This manuscript is a resubmission of an earlier submission. The following is a list of the peer review reports and author responses from that submission.

Round 1

Reviewer 1 Report

General Comments:
The paper describes designing of a liquid oral pediatric formulation of ranitidine hydrochloride. The concept is fine but the paper discloses preparation and assessment of only one liquid oral formulation which is insufficient looking at the title of the paper. I wonder how authors reached this optimized formulation, especially the amounts of sweeteners used and their synergistic action. Taste masking, an important attribute of pediatric formulation, was not evaluated with stability samples. As some degradation of ranitidine was evident through chromatographic studies, taste may be compromised after one month of storage. Further study for identification of degradants seems essential to confirm the identity of degradants. Limitations of the study have not been provided anywhere in the article. Therefore, the article in present form seems a very early report. In the present form, the article may be sent back to authors and a resubmission after necessary add-on studies is recommended. Specific comments are as follows:

Specific Comments
Line 128 Table 2: Only one formulation is prepared. No attempt is evident to vary the amounts of sweeteners to optimize taste masking.Formulation development is an essential step. Explain synergistic action of sweeteners at amounts used in the formulation.
Line 128 Table 2: No units have been provided for components in the formulation.
Line 143: Please add compositions of polymeric cross-sensitive sensor membranes and their purpose in the taste sensor. 
Line 160 Correct "HCL" as "HCl"
Line 222 and Table 3: Linear calibration curve was constructed below LOQ. Please clarify.
Line 236: No detail for Multivariate E-tongue classification models (LDA-SA-E-tongue models) is provided in the paper.
Line 237: What are LOO-CV procedures? Please add details
Line 358: Expand PCDPD.

Reviewer 2 Report

Overcoming challenges in pediatric formulation with a patient- 2 centric design approach

General comments: The article is interesting, bringing also general information about the pediatric products and its challenges. General gramar and english language should be reviewed. Improvement suggestions are mentioned below:

Abstract:

-Consider using the verb was in “The purpose of the present study is...”

Introduction:

-TPP (Table 1) should be shown in the methodology or results, instead of in article introduction. Also, comment column of such table needs more references. Only the “flavor” was established based on a cited reference.

- Line 52 – Change “Design” for “To design” in the beginning of the sentence.

- Include a coma after “With respect to selection of excipients and vehicle several” and after “To successfully design an oral medicine for pediatric”

- Include “patients” or “population” after pediatric, line 78.

- Insert comas before and after “thus”, line 89

- Such paragraph is too short: “Patient centric drug product design considers both the target drug product and the patient profile and integrate this insight by establishing a Target Product profile (TPP).” Please combine it with another paragraph.

- Change “work” for “investigation” or “study” in the entire text.

- The novelty of the investigation is not clear. Please emphasize it in the introduction, bringing  information about similar articles in the literature and also information about ranitidine oral solution available on the market.

- Change “is to design” for “was to design”, line 103.

Materials and Methods:

- 2.1 should be written as a single paragraph. Change conductance for conductivity.

- Table 3 should be placed in the results, since validating the hplc methodology was included in the methodology.

- The stability of the aqueous ranitidine solution and of the syrup should have been performed. It would help discussing the results.

Results and Discussion:

- Please review the writing of this sentence: “To control consisting of an unopened recipients stored at 4ºC and RT were included for comparison”. Line 265

- Table 4 and 5 should be combined in one, to make reading easier. Consider also including the microbiological study in such table.

- Please check the writing of the sentence: “The content of Ra-HCl was higher than 98% for all test conditions (Table 5) which confirm the drug stability in the presence of the excipients used and solution pH and ensures a beyond use date of at least 30 days.”

- Line 279 please change “30 days storage” for “30 days of storage”

- Regarding stability, for a better understanding, the name of the formulation should be kept always the same, for exemple authors call “open daily” or “in use” depending on the part of the article.

- Considering the degradation products formed, the authors should only consider the product stable at 4oC for 30 days without opening. This should be discussed along with the TPP. Also, the authors should discuss more deeply the degradation products probably formed. Are they described in the literature? Are they toxic? ...

- The acronym “PCDPD” was used without explaining what it means.

Reviewer 3 Report

The title is too vague and over selling a very simple study on one extemporaneous preparation of an API that is not currently available in many countries as it has been discontinued as a precaution because it may contain a small amount of an impurity that has been linked to an increased risk of cancer in animals. It's not yet known whether it will be available again in future.

There is no justification of the 'challenges' it presents in terms of dose requirement as a function of age, need for dose flexibility, palatability and stability issues. 

There is no reference to the other ranitine extemporaneous formulations described in the literature.

The method is not validated to measure stability and the results of the lab made e tongue are difficult to understand outside of any in vivo taste data context.

There is a lot emphasis on the importance of justifying the excipients used yet it has not been done for, for example, the intense sweeteners.

The conclusion propose the formula as a cheap taste masking approach that could be used for other drugs but it shows the lack of understanding of the challenges different drugs with different level of bitterness can present.